# Gastric Cancer Extracellular Vesicles Tune the Migration and Invasion of Epithelial and Mesenchymal Cells in a Histotype-Dependent Manner

**DOI:** 10.3390/ijms20112608

**Published:** 2019-05-28

**Authors:** Sara Rocha, Sara Pinto Teles, Mafalda Azevedo, Patrícia Oliveira, Joana Carvalho, Carla Oliveira

**Affiliations:** 1i3S–Instituto de Investigação e Inovação em Saúde, Universidade do Porto, 4200-135 Porto, Portugal; srocha@ipatimup.pt (S.R.); steles@ipatimup.pt (S.P.T.); poliveira@ipatimup.pt (P.O.); 2Ipatimup—Institute of Molecular Pathology and Immunology of University of Porto, 4200-135 Porto, Portugal; 3ICBAS—Instituto de Ciências Biomédicas Abel Salazar, Universidade do Porto, 4050-313 Porto, Portugal; azevedom@mskcc.org; 4Program in Developmental Biology, Sloan Kettering Institute, Memorial Sloan Kettering Cancer Center, New York, NY 10065, USA; 5Department Pathology, Faculty of Medicine, University of Porto, 4200-319 Porto, Portugal

**Keywords:** extracellular vesicles, gastric cancer, epithelial-to-mesenchymal transition, invasion, migration

## Abstract

Extracellular vesicles (EVs) secreted by tumor cells modulate recipient cells’ behavior, but their effects in normal cells from the tumor microenvironment remain poorly known. In this study, we dissected the functional impact of gastric cancer cell-derived EVs (GC-EVs), representative of distinct GC histotypes, on the behavior of normal isogenic epithelial and mesenchymal cells. GC-EVs were isolated by differential centrifugation and characterized by transmission electron microscopy, nanoparticle tracking analysis, and imaging flow-cytometry. Epithelial and mesenchymal cells were challenged with GC-EVs and submitted to proliferation, migration, and invasion assays. Expression of epithelial and mesenchymal markers was followed by immunofluorescence and flow-cytometry. Our results indicated that GC-EVs secreted by diffuse-type cancer cells decrease the migration of recipient cells. This effect was more prominent and persistent for mesenchymal recipient cells, which also increased Fibronectin expression in response to EVs. GC-EVs secreted by cancer cells derived from tumors with an intestinal component increased invasion of recipient epithelial cells, without changes in EMT markers. In summary, this study demonstrated that GC-EVs modulate the migration and invasion of epithelial and mesenchymal cells from the tumor microenvironment, in a histotype-dependent manner, highlighting new features of intestinal and diffuse-type GC cells, which may help explaining differential metastasis patterns and aggressiveness of GC histotypes.

## 1. Introduction

Extracellular vesicles (EVs) are crucial mediators of the “dialog” between tumor and stromal cells at local and distant microenvironments [1]. These secreted vesicles carry a myriad of pro-tumoral and pro-metastatic factors that qualify them to orchestrate the invasion-metastasis cascade by reprogramming both tumor and stromal cells [2,3,4]. Several studies have shown that EVs participate in the initial steps of tumor cell invasion by, for instance, inducing a transdifferentiating program known as epithelial to mesenchymal transition (EMT) [5]. Particularly, EVs secreted by tumor cells enclose a pro-EMT cargo that enhances cell migration and invasion via downregulation of epithelial (e.g., E-cadherin and ZO-1) and upregulation of mesenchymal (e.g., Vimentin) markers [6,7,8]. This EMT program can also be triggered by EVs secreted by stromal cells, such as fibroblasts [9,10]. Indeed, breast cancer-associated fibroblasts secrete EVs that promote breast cancer cell protrusive activity and motility through the Wnt-planar cell polarity signaling [11].

Particularly in gastric cancer (GC), several studies have explored the role of EVs in the crosstalk between tumors and several types of stromal cells [12]. GC-derived EVs (GC-EVs) have been shown to sustain tumor growth by triggering the differentiation of human umbilical cord-derived mesenchymal stem cells (MSCs) into carcinoma-associated fibroblasts (CAFs) via Transforming Growth Factor beta (TGFβ), or reprogram fibroblasts and pericytes into CAFs by transferring miR-27a and BMP, respectively [13,14]. GC-EVs may also affect the immunomodulatory functions of MSCs through Nuclear Factor Kappa B (NF-κB) signaling [15,16] and induce the expression of pro-inflammatory factors in macrophages or of autophagy in neutrophils [17,18], thus promoting tumor cell proliferation and migration. Recently, it was demonstrated that these are bi-directional communications such as CAF-derived EVs induce scirrhous-type GC cell migration and invasion through MMP2 activation [19], and MSCs-derived EVs induce EMT and stemness of GC cells, further promoting cancer cell growth and migration [20]. As reported in other cancer models [4,21], GC-EVs have also been implicated in the remodeling of the pre-metastatic microenvironment. Zhang et al. showed that EGFR-containing GC-EVs facilitate liver-specific metastasis through the suppression of miR-26a/b expression and activation of HGF on liver stromal cells [22]. Further, two independent studies have provided insights into the mechanism for peritoneal metastasis of GC. Whilst Deng et al. demonstrated that GC-EVs facilitate peritoneal metastasis through mesothelial-to-mesenchymal transition (MMT) of peritoneal mesothelial cells via upregulation of p-ERK, Li et al. found that this role is mediated by miR-21-5p-containing GC-EVs, which promote cancer dissemination by targeting SMAD7 [23,24]. 

Notwithstanding, it remains to be elucidated how GC cells modulate not only non-transformed epithelial cells present in the tumor microenvironment but also mesenchymal cells arising in the context of epithelial-to-mesenchymal transition in often-inflamed tumor microenvironments. Further, no evidence exists about the role of EVs in mediating the first steps of tumor cell migration and invasion of intestinal- and diffuse-type GC, the main histological types of this disease. Herein, we sought to dissect the functional impact of EVs, isolated from GC cell lines, representative of intestinal and diffuse histotypes, on the behavior of normal epithelial and mesenchymal cells. 

## 2. Results

To explore whether EVs released by GC cells modulate non-transformed epithelial and mesenchymal cells located in the tumor microenvironment, we designed the following study according to the scheme presented in Figure 1. Selected GC cell lines are representative of distinct histotypes of GC. The MKN74 GC cell line represented the intestinal-type GC, Kato III and IPA220 cells presented characteristics resembling the diffuse-type GC, and MKN45 harbored both intestinal and diffuse components (Appendix A) [25,26].

### 2.1. TGFβ1 Induces EMT in a Non-Tumorigenic Epithelial Cell Line

First, we established and characterized a human TGFβ-induced EMT cell recipient model. As there is no non-tumorigenic gastric cell line currently available, we induced EMT in MCF10A, a non-tumorigenic breast epithelial cell line.

By adapting a previously described approach [27] and using TGFβ1 as a classical inducer of EMT, we derived, from the same genotypic background, two distinct phenotypic cell types: epithelial and mesenchymal. Whilst epithelial cells presented cobblestone morphology, the mesenchymal counterpart acquired an elongated spindle-shape (fibroblastic-like) morphology and lost cell–cell adhesion (Figure 2A). To confirm whether the observed phenotypic alterations were due to EMT, we analyzed the expression of well-described epithelial and mesenchymal markers, both at the RNA and protein levels. Upon EMT, mesenchymal cells displayed a significant decrease of epithelial *CDH1* and *OCLN* and a substantial increase of mesenchymal *CDH2*, *VIM,* and *FN* mRNA expression (Figure 2B). These alterations were also detected at the protein level, where mesenchymal cells lost E-cadherin and gained Fibronectin in comparison to epithelial cells (Figure 2C).

Overall, these results point to the occurrence of EMT, and set the ground for the usage of these two isogenic epithelial and mesenchymal cell lines as recipients of GC-EVs. 

### 2.2. Distinct GC Cell Lines Secrete EVs with Similar Physical and Biochemical Properties

Next, we isolated and characterized EVs secreted by four GC cell lines. We observed that all GC cell lines presented a high percentage of viable cells at the time of conditioned media collection and isolation of EVs (Appendix A), thus reducing the potential contamination of EV preparations by apoptotic bodies [28]. The harvested EVs were analyzed by transmission electron microscopy (TEM), nanoparticle tracking analysis (NTA), and imaging flow cytometry (Figure 3). TEM showed that vesicles recovered at 100 K were of the size generally assigned to exosomes (Figure 3A) [29], but in all cell lines except MKN74, a fair concentration of larger EVs (~200 nm) were also present. These results were also confirmed by NTA, which revealed a mean size of 111, 116, 121, and 127 nm, for MKN74-, MKN45-, Kato III-, and IPA220-EVs, respectively (Figure 3B). Moreover, imaging flow cytometry detected commonly associated exosomal markers CD9, CD81, and Flotillin-1 in EVs secreted by the four distinct GC cell lines (Figure 3C) [30]. However, EVs from all four GC cell lines seem to have made different contributions of these markers, with IPA220 showing equivalently high representation of CD9 and CD81, and MKN74 showing the lowest representation of both markers (Figure 3C). Given the differences in these EV-associated markers, it is likely that the cargo of these EVs may also be different.

Despite the later differences, globally our results seem to indicate that EVs secreted by distinct types of GC cell lines present relatively similar physical and biochemical properties. 

### 2.3. GC-EVs Do Not Modulate E-Cadherin Expression at the Cell Membrane

Since E-cadherin is a classical EMT marker, we first checked whether GC-EVs modulate its expression and localization in recipient cells. We treated epithelial and mesenchymal recipient cells with EVs from the four donor GC cell lines and, upon 24 h of treatment, we analyzed the expression and localization of E-cadherin by flow cytometry (Figure 4). As expected, the non-treated (control) mesenchymal cells presented a significantly reduced percentage of membrane E-cadherin-positive cells, in comparison with non-treated (control) epithelial cells (*p* ≤ 0.0001, Figure 4A). When epithelial and mesenchymal cells were treated with the four distinct GC-EVs, neither the percentage of E-cadherin-positive cells at the membrane (Figure 4A) nor the intensity of surface E-cadherin expression (Figure 4B) varied between treated and non-treated cells. 

These results suggested that GC-EVs do not interfere with the levels of E-cadherin expressed at the membrane of both epithelial and mesenchymal recipient cells. 

### 2.4. GC-EVs Impair Migration and Invasion of Epithelial and Mesenchymal Cells 

Our results suggest that the main hallmark molecule of EMT—E-cadherin—remains unaffected when near-normal epithelial and mesenchymal cells are exposed to GC-EVs. However, growing evidences demonstrate that cell phenotypes, such as migration and invasion, may be modulated even in cancer cells expressing E-cadherin [31]. For this reason, we assessed whether cell migration, proliferation, and invasion of epithelial and mesenchymal recipient cells could be modulated in the presence of GC-EVs.

We start by examining whether GC-EVs influence the migration rate and pattern of the different recipient cell types, by performing wound-healing assays followed by time-lapse bright-field microscopy. In the absence of GC-EVs, epithelial and mesenchymal cells (controls) displayed a similar migration rate, although with distinct patterns (Figure 5A,B). Whilst epithelial cells migrated in a collective manner, mesenchymal cells displayed a single cell migration behavior (Figure 5A). Then, we analyzed the effects of GC-EVs in recipient cells. We verified that EVs secreted by Kato III and IPA220 GC cells impaired significantly the migration of both epithelial and mesenchymal cells, by decreasing their rate, but without affecting migration patterns (Figure 5A,B). To overcome variability among experiments, we normalized the data of treated cells to the respective control cells, considering each time-point (Figure 5C). Specifically, we found that IPA220-EVs led to a reduction in the migration rate of epithelial cells in the first 4 h, which was recovered at later time-points (Figure 5C, left). Regarding the migration rate of mesenchymal cells, both Kato III- and IPA220-EVs led to a more permanent reduction in the first 4 h that was maintained during 12 and 20 h, respectively (Figure 5C, right). Strikingly, mesenchymal cells challenged with IPA220-EVs were not able to close the wound area in 20 h comparing with non-treated mesenchymal cells (Figure 5A,B). 

To further determine whether distinct levels of proliferation underlined changes in migration rates, we measured EdU incorporation into newly synthetized DNA by flow cytometry. We observed that, in the absence of GC-EVs, mesenchymal cells proliferate less than epithelial cells, although with no statistical significance (18.7% vs. 29.7% of EdU positive cells, Appendix A). When challenged with GC-EVs, both epithelial and mesenchymal cells maintained the proliferation levels of the corresponding non-treated counterparts, meaning that these GC-EVs had no effect on EMT-related proliferation (Appendix A). 

Taken together, these results pinpointed that EVs from specific GC cell lines are able to interfere with the migration rates of mesenchymal cells, mainly, without affecting their proliferation levels. Furthermore, in light of the above results on E-cadherin membranous expression, these changes in migration of epithelial and mesenchymal recipient cells may even occur independently of their E-cadherin expression. Importantly, the greater impacts in migration were achieved when recipient cells were treated with EVs from cell lines deriving from diffuse-type GC (poorly differentiated).

Next, we evaluated whether EVs from the different types of GC cells differentially modulate the invasion of epithelial and mesenchymal cells (Figure 6). We observed that epithelial cells challenged with MKN74-, MKN45-, and IPA220-EVs tend to be more invasive (fold change: 2.37, 3.38, and 1.95, respectively) than the non-treated epithelial counterpart; however, this effect was only statistically significant for MKN45-EVs (adjusted *p*-value 0.0143, Figure 6A). In contrast, mesenchymal cells became less invasive upon treatments with MKN45-, Kato III-, and IPA220-EVs (fold change: 0.58, 0.36, and 0.55, respectively) in comparison with non-treated mesenchymal cells (Figure 6B). Despite the lack of statistical significance observed for all conditions, it was clear that Kato III-EVs had a stronger impact on invasion of mesenchymal cells than MKN45- and IPA220-EVs. 

To check whether the impact of GC-EVs on the invasion capacity of epithelial and mesenchymal cells was accompanied by differences in the expression levels of EMT markers, we performed E-cadherin and Fibronectin immunocytochemistry 48 h after EVs treatment, i.e., at the endpoint of the invasion assay. Concerning the epithelial marker E-cadherin, no differences were observed upon challenging of epithelial and mesenchymal cells with GC-EVs comparing with the respective non-treated counterparts (Figure 6C). Interestingly, we observed that the reduced invasion capacity of mesenchymal cells upon treatment with Kato III- and IPA220-EVs was accompanied by an increase of Fibronectin expression (Figure 6C). 

Invasion of epithelial and mesenchymal recipient cells seems to be correlated with the histotype from which GC cell lines have been derived. Indeed, EVs from MKN74 and MKN45 cell lines, which display an intestinal component, promote invasion in epithelial recipient cells but do not affect mesenchymal recipient cells. In contrast, EVs from diffuse-type GC cell lines (Kato III and IPA220) do not influence invasion of epithelial cells but demonstrate a trend towards decreasing the invasion of mesenchymal recipient cells.

Collectively, these findings indicated that GC-EVs modulate the behavior of epithelial and mesenchymal recipient cells by impacting their cell migration, their invasion, and the expression of EMT markers. In particular, our results suggest that epithelial and mesenchymal recipient cells respond differentially to the same GC-EVs. Moreover, the effects of GC-EVs over recipient cells may depend on the histotypes of the EV-donor cells.

## 3. Discussion 

Communication between tumor cells and the associated stroma plays a key role in driving tumor progression, but how GC cells modify the environment for their own benefit is less clear. In the present study, we hypothesized that EVs secreted by GC cells are able to subjugate normal cells within the tumor stroma to participate in cancer progression. We tested this hypothesis by focusing on two different recipient stromal cell types that often co-exist in the vicinity of cancer cells—normal epithelial and mesenchymal cells. Normal epithelial cells represent the adjacent normal cells surrounding the tumor, while mesenchymal cells represent epithelial cells exposed to transdifferentiating agents, such as inflammatory factors (e.g., TGFβ), often arising during EMT [32]. To this end, we established and characterized a human TGFβ1-induced-EMT cell model to obtain both epithelial and mesenchymal recipient cells for further treatment with distinct GC-EVs. We chose to use the non-tumorigenic breast epithelial cell line (MCF10A) as a recipient cell type to induce EMT, given that (1) non-tumorigenic gastric cell lines are currently not available; (2) primary epithelial gastric cells hardly proliferate and maintain viability in vitro, which makes them incompatible with our experimental design; and (3) breast epithelial cancers, but rarely other cancers from the digestive system, have been identified in the tumour spectrum of GC-associated syndromes, such as hereditary diffuse gastric cancer [33]. This last evidence supports breast epithelia as a tissue mimicking gastric epithelia regarding the tumorigenic process triggered by germline alterations. Altogether, the normal epithelial breast cell line (MCF10A), although not ideal, seems like an acceptable model to be used as a recipient cell line for GC-EVs. This recipient model underwent EMT when treated with TGFβ1, as demonstrated by cell morphological alterations and the differential expression of classical epithelial and mesenchymal markers [34]. We then chose four GC cell lines, representative of distinct GC histological types and differentiation patterns, to dissect the functional impact of GC-EVs on epithelial and mesenchymal cells. To the best of our knowledge, this is the first study addressing the impact of GC-EVs from different histological types over otherwise normal epithelial and mesenchymal stromal cells. 

GC is histologically classified into two main subtypes—intestinal and diffuse—which are recognized by distinct morphological features, differentiation patterns, and clear-cut differences on the metastatic pattern [35,36]. Intestinal-type GC is characterized by well-differentiated glandular structures and metastasizes, mainly to the liver, whilst diffuse-type GC consists of individually neoplastic cells that infiltrate the stomach wall and usually forms peritoneal and lymph node metastasis [37]. Notwithstanding, the two histotypes share several sequential and interrelated events, such as local invasion into the surrounding tumor-associated stroma [38]. 

The most striking result of this study shows that as soon as epithelial and mesenchymal recipient cells interact with GC-EVs derived from diffuse-type GC cell lines (Kato III and IPA220), they start migrating less. This effect is equivalent for epithelial and mesenchymal recipient cells at early time-points, but persists longer in mesenchymal recipient cells. Interestingly, Kato III- and IPA220-EVs also induced the highest expression of Fibronectin in mesenchymal cells. These findings together may suggest that diffuse-type GC cells make use of EVs to render the microenvironment permissive for their further colonization and growth at preferred metastatic sites, such as the peritoneum. Actually, our findings may resemble some events occurring in peritoneal metastasis. As such, it has been demonstrated that GC-EVs facilitate peritoneal metastasis by disrupting the mesothelial barrier through induction of mesothelial to mesenchymal transition [23]. Additionally, it has been demonstrated that GC-EVs can stimulate the expression of adhesion-related molecules, such as fibronectin 1 and laminin gamma 1, in mesothelial cells [39]. This alteration fosters the adhesion of GC cells to mesothelial cells, thus creating a favorable scenario for dissemination of GC cells and consequent metastasis. From this perspective, it is sensible to hypothesize that GC-EVs derived from diffuse-type GC cell lines act particularly over mesenchymal cells in the tumor vicinity, and eventually adhere to them through fibronectin (yet to be seen) towards peritoneal colonization.

We also found that EVs derived from GC cells with an intestinal-type component were able to promote invasion of normal epithelial cells, with the most pronounced effect observed for EVs secreted by a GC cell line displaying both diffuse and intestinal morphologies (MKN45). Interestingly, this invasive phenotype was not accompanied by an increased migration. As such, one may hypothesize that the cargo of MKN45-EVs is more important for the remodeling of the basement membrane than for the direct movement of epithelial cells. Additionally, E-cadherin and Fibronectin expression levels remained unchanged in recipient cells, suggesting that other epithelial/mesenchymal markers may be responsible for promoting the observed invasive phenotype. 

Our results indicate that GC-EVs secreted by cancer cells, derived from diffuse-type tumors, increase migration of recipient mesenchymal cells, while GC-EVs secreted by cancer cells derived from tumors with an intestinal component increase invasion of recipient epithelial cells in the tumor microenvironment. These results suggest that GC-EVs representative of different GC histotypes may have distinct cargo and membrane compositions, which may influence their uptake by recipient cells. On the other hand, epithelial (E) and mesenchymal (M) cells may also uptake and perceive GC-EVs differently. In fact, potential distinctive uptakes of GC-EVs by recipient cells may represent how effectively these EVs may convey tumor messages to their microenvironment. Several studies on different cancer models have demonstrated that EVs may indeed deliver signals able to induce EMT, thus initiating the invasion-metastasis cascade. For instance, Franzen et al. proved that bladder cancer-derived EVs increased the expression of several mesenchymal markers (including α-smooth muscle actin, S100A4, and snail), as well as the motility and invasiveness of normal recipient uroepithelial cells [6]. These phenotypes were also detected when lung-cancer EVs derived from highly metastatic cells were applied to normal epithelial cells [8]. Specifically, these lung-cancer EVs have been found to increase the expression of Vimentin and N-cadherin and decrease E-cadherin and ZO-1. Additionally, breast and colon cancer-EVs containing amphiregulin boosted invasiveness of both epithelial and breast cancer cells [2]. These and others studies highlighted that EVs isolated from different cancer types induce similar phenotypes on normal epithelial cells, although using distinct players and molecular pathways [40,41]. 

In summary, the present study demonstrated that GC-EVs modulate the migration and invasion of epithelial and mesenchymal cells in a histotype-dependent manner, highlighting new features of intestinal and diffuse-type GC cells, which may help explaining the metastasis pattern, aggressive phenotypes, and prognosis of GC histological types. 

## 4. Materials and Methods 

### 4.1. Establishment and Characterization of Epithelial-to-Mesenchymal Transition EMT Recipient Model 

#### 4.1.1. Treatment of MCF10A Epithelial Cell Line with TGFβ1 

A human TGFβ1-induced-EMT cell model was established as previously described [27]. The non-tumorigenic epithelial cell line MCF10A (E-cells) was cultured in D-MEM/F12-GlutamaxTM (Thermo Fisher Scientific, Waltham, MA, USA) supplemented with horse serum (5%, Thermo Fisher Scientific), penicillin-streptomycin (PS, 1%, Thermo Fisher Scientific), recombinant human insulin (5 μg/mL, Sigma-Aldrich, St. Louis, MO, USA), recombinant human epidermal growth factor (20 ng/mL, Sigma-Aldrich), cholera toxin (20 ng/mL, Sigma-Aldrich), and hydrocortisone (500 ng/mL, Sigma-Aldrich). Mesenchymal cells (M cells) were obtained by treating 6 × 10^4^ epithelial cells (E cells) during seven days with the abovementioned culture medium supplemented with transforming growth factor-β1 (TGFβ1, 8 ng/mL, Sigma- Aldrich).

#### 4.1.2. Co- Immunofluorescence of E-Cadherin and Fibronectin 

MCF10A cells (E and M) were co-immunostained for E-cadherin (1:100, 24E10 Cell Signaling, Danvers, MA, USA) and Fibronectin (1:100, clone 2755-8, Santa Cruz, Dallas, Texas, TX, USA). Briefly, cells were fixed in 4% PFA (20 min), followed by treatment with NH_4_Cl (50 × 10^−3^ M, 10 min, Merck, Darmstadt, Germany) and Triton X-100 (0.2%, Sigma-Aldrich, 5 min) and blocking with BSA (Bovine Serum Albumin, 5%, 30 min, NZYTech, Lisbon, Portugal). Upon overnight (ON) incubation at 4 °C with a mixture of rabbit anti-E-cadherin and mouse anti-Fibronectin primary antibodies, cells were co-incubated with secondary antibodies anti-rabbit Alexa 594 and anti-mouse Alexa 488 (1:500, 1 h, Thermo Fisher Scientific) at room temperature (RT). Images were taken with a Zeiss Imager.Z1, AxioCam MRm (Zeiss, Oberkochen, Germany).

#### 4.1.3. RNA Expression Quantification of Epithelial and Mesenchymal Markers 

RNA was extracted from three biological replicas of E and M cells following instructions of the mirVana miRNA Isolation Kit (Thermo Fisher Scientific), and reversed transcribed (1000 ng) to cDNA with Superscript II Reverse Transcriptase and random hexamer primers (Thermo Fisher Scientific). Quantitative real time-PCR (qRT-PCR) was carried out in triplicates for the epithelial (*CDH1* and *OCLN*) and mesenchymal (*CDH2*, *VIM* and *FN*) target genes and for the endogenous control *GAPDH* using PrimeTime-qPCR assays (IDT, Hs.PT.53. 2388193; Hs.PT.49.14927371; Hs.PT.58.45367437; Hs.PT.47.14705389; Hs.PT.58. 40986315; and Hs.PT.51.1940505) on a 7500 Fast Real-Time PCR system (Thermo Fisher Scientific). Data was analyzed by the comparative 2^−ΔΔ*C*T^ method [42]. For all data comparisons, the Unpaired *t* test with Welch’s correction was used.

### 4.2. Characterization of GC-EVs 

#### 4.2.1. GC Cell Culture and Viability 

Human GC cell lines MKN74, MKN45, Kato III (ATCC), and IPA220 (established at Ipatimup [26]) were cultured in RPMI 1640 medium (Thermo Fisher Scientific) supplemented with 10% fetal bovine serum (FBS, Biowest, Nuaillé, France) and 1% PS (Penicillin-Streptomycin, Thermo Fisher Scientific) at 37 °C in 5% CO_2_ humidified atmosphere. GC cells were seeded in T175 flasks and maintained with the abovementioned medium until a confluence of 60–70% was reached. GC cells were further cultured in EV-depleted medium during 48 h. This medium was obtained as previously described [43]. Briefly, RPMI medium supplemented with 20% FBS and 1% PS was centrifuged ON at 100,000× g, filtered, and diluted in the same amount of RPMI medium supplemented with 1% PS and without FBS. EV-depleted medium was further submitted to TEM to ascertain that the depletion was correctly performed.

GC cell viability was measured using annexin V-FITC and PI double staining followed by flow cytometry, as described [43]. 

#### 4.2.2. EV Isolation by Differential Centrifugation

GC-EVs were isolated from the conditioned media of GC cell cultures, growing in EV-depleted medium, by differential centrifugation, as previously described [43]. Briefly, the conditioned media of each GC cell line was submitted to two sequential centrifugation steps (at 300× g for 10 min and 2000× g for 20 min) followed by a filtration step. The filtered supernatant was centrifuged at 100,000× g, 4 °C, in a SW32 rotor (Beckman Coulter, Fullerton, CA, USA) for 4 h to pellet EVs, and was further washed in PBS without Ca^2+^ and Mg^2+^ (Thermo Fisher Scientific) and centrifuged at 100,000× g, 4 °C, for 2 h. Each pellet containing GC-EVs was resuspended in an appropriate volume of 0.9% NaCl, in accordance with downstream applications.

#### 4.2.3. EV Morphology by Transmission Electron Microscopy (TEM)

GC-EVs were negatively stained and visualized by Transmission Electron Microscopy. Briefly, EV preparations were added to Formvar-carbon-coated grids and incubated for 1 min at RT in the dark, to allow adsorption to the grid. EVs were further stained with 5% uranyl acetate during 1 min and examined with a JEM 1400 electron microscope (JEOL, Tokyo, Japan). Images were recorded using a SC1000 Orius CCD camera (Gatan, Pleasanton, CA, USA).

#### 4.2.4. EV Characterization by Nanoparticle Tracking Analysis (NTA)

The size and concentration of GC-EVs were measured by NTA, using the NanoSight NS300 (Malvern, Worcestershire, UK) as described [43]. Mode and mean size were calculated from the three videos per biological replicate. Results represent the mean ± standard deviation of at least 14 biological replicates of each cell line.

#### 4.2.5. EV Characterization by Imaging Flow Cytometry 

GC-EVs were analyzed for the expression of CD9, CD81, and Flotillin-1 using imaging flow cytometry (ImageStreamX Amnis Corporation, Seattle, WA, USA) as described [43]. Briefly, EVs were coupled to previously sonicated aldehyde/sulfate latex beads (1 × 10^9^ particles determined by NTA per 3 μL of beads) for 1 h at RT with agitation followed by ON incubation at 4 °C and blocking with glycine solution with agitation (100 × 10^−3^ M 30 min; Merck). Next, EVs–beads were immunostained with mouse monoclonal anti-CD9 (sc-59140, Santa Cruz), anti-CD81 (sc-23962, Santa Cruz), and rabbit polyclonal anti-Flotillin-1 (sc-25506, Santa Cruz) for 1 h, followed by incubation with goat anti-rabbit/anti-mouse Alexa Fluor 488 secondary antibodies (Invitrogen, 30 min) and analysis by imaging flow cytometry. For each sample, 100 000 events were acquired at a 40× magnification. Fluorescence of the stained EVs–beads was excited with a 488 argon laser and collected on channel 2 (505–560 nm). A 745 nm laser was activated for side scatter and collected on channel 6, and bright-field images with adjusted intensity were collected on channel 1. Data analysis was performed using the IDEAS software (Amnis Corporation).

### 4.3. Functional Assays with Recipient Cells Treated with GC-EVs

#### 4.3.1. Treatment of Epithelial and Mesenchymal Cells with GC-EVs

After establishing M cells by treating E-cells (60,000 cells) for seven days with TGFβ1 (as abovementioned), both E and M cells were seeded in 6-well plates (100,000 cells/well) and maintained during 24 h in normal culture medium. Then, the normal culture medium was replaced by EV-depleted medium and cells were treated with 1 × 10^9^ of GC-EVs (TGFβ1 was maintained in M cultures). After 24 h of treatment, E and M cells were collected and submitted to different assays. 

#### 4.3.2. Flow Cytometry Analysis of E-Cadherin

Flow cytometry was performed as described [44]. Briefly, E and M cells (treated and non-treated with GC-EVs) were detached with Versene (Thermo Fisher Scientific) and resuspended in ice-cold PBS with 0.05 mg/mL CaCl_2_. For all conditions, cells were centrifuged at 1500 rpm for 5 min at 4 °C and washed in PBS with 0.05 mg/mL CaCl_2_ and 3% BSA. Cells were then incubated for 1 h with mouse anti-E-cadherin primary antibody HECD-1 (1:50 dilution, Thermo Fisher Scientific), washed twice, and incubated with secondary antibody anti-mouse Alexa 488 (1:250 dilution, Thermo Fisher Scientific) in the dark for 1 h. At least 2 × 10^4^ single cells were acquired using FACS Canto II cytometer, and further analyzed using FlowJo software (Ashland, Oregon, OR, USA). Results represent the mean ± standard deviation of three biological replicates. Data was analyzed by Two-way ANOVA with Tukey’s multiple comparisons test.

#### 4.3.3. Wound-Healing Migration Assay

Wound-healing assays of E and M cells (treated and non-treated with GC-EVs) were performed using µ-Plate 24 well with culture-inserts (Ibidi) and time-lapse microscopy. Briefly, 4 × 10^4^ cells were seeded in each of the two wells of the culture insert and grown to confluence for, approximately, 24 h. After this period, the culture insert was removed, cells were washed with PBS, and fresh EV-depleted medium supplemented with GC-EVs (5 × 10^8^) was added. Bright field images of the wounds were photographed immediately after removing the insert (0 h) and again at 4 h, 8 h, 10 h, 12 h, 16 h, and 20 h using Leica DMI6000 (Wetzlar, Germany). Five images were automatically captured per well per time point to completely image each wound. Cells were maintained at 37 °C in 5% CO_2_ humidified atmosphere during the time-lapse imaging process. The total area of the wound was measured at each time point using ImageJ software (MRI_Wound_Healing_Tool). Results represent the mean ± standard deviation of three biological replicates. Data was analyzed with a two-way ANOVA with Dunnett’s multiple comparison test.

#### 4.3.4. Proliferation Assay

Proliferation assays of E and M cells (treated and non-treated with GC-EVs) were performed following the instructions of the Click-iT™ Plus EdU Pacific Blue Flow Cytometry Assay Kit (Thermo Fisher Scientific). At least 2 × 10^4^ single cells were acquired using FACS Canto II cytometer, and further analyzed using FlowJo software. Results represent the mean ± standard deviation of three biological replicates.

#### 4.3.5. Matrigel Invasion Assay

Invasion assays of E and M cells (treated and non-treated with GC-EVs) were performed using Corning BioCoat Matrigel Invasion Chambers (Thermo Fisher Scientific). Briefly, after 24 h of treatment with GC-EVs as abovementioned, 5 × 10^4^ of E and M cells were seeded in the upper compartment and incubated for 24 h at 37 °C in 5% CO_2_. Filters were then washed in PBS and fixed in ice-cold methanol for 10 min. After removal of non-invasive cells adherent to matrigel with a prewet cotton swab, each filter was washed and mounted in glass coverslips with Vectashield/DAPI. A mosaic of the entire filter was obtained with Axiovert 200 M (Zeiss), and the total number of invasive nuclei was counted using ImageJ software. Results represent the mean ± standard deviation of three biological replicates. Data was analyzed with a one-way ANOVA with Kruskal-Wallis test.

#### 4.3.6. Co-Immunofluorescence of E-Cadherin and Fibronectin

E and M cells (treated and non-treated with GC-EVs) were seeded in coverslips and grown during 24 h. The choice of this time-point was based on the one adopted for the invasion assay. After this, co-immunostaining for E-cadherin and Fibronectin was performed according to the protocol described above. Images were acquired with Zeiss Imager.Z1, AxioCam MRm, or Leica DMI6000.

### 4.4. Statistical Analysis

The Unpaired *t* test with Welch’s correction was used for comparisons of RNA expression levels of epithelial and mesenchymal markers. Flow-cytometry and wound-healing data were analyzed by a Two-way ANOVA with Tukey’s and Dunnett’s multiple comparisons tests, respectively. Matrigel invasion data was analyzed with a one-way ANOVA with a Kruskal–Wallis test.

## Figures and Tables

**Figure 1 ijms-20-02608-f001:**
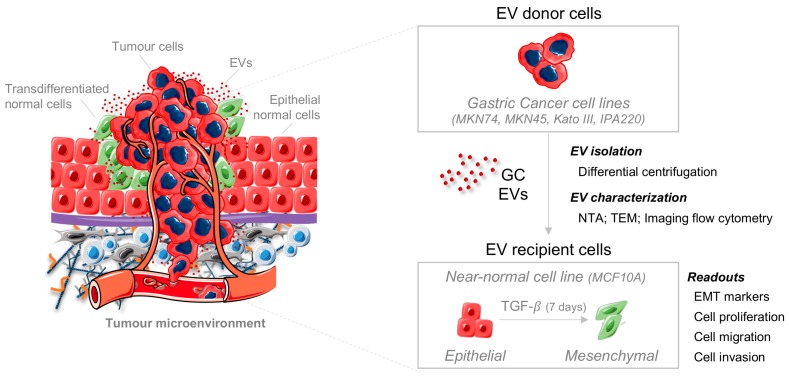
Scheme illustrating the experimental design of the study. Four distinct gastric cancer (GC) cell lines (MKN74, MKN45, Kato III, and IPA220) were used as donor cells of extracellular vesicles (EVs), which were isolated by differential centrifugation and characterized by transmission electron microscopy (TEM), nanoparticle tracking analysis (NTA), and imaging flow cytometry. Epithelial (red) and mesenchymal (green) cells were used as recipients of GC-EVs and submitted to distinct functional assays.

**Figure 2 ijms-20-02608-f002:**
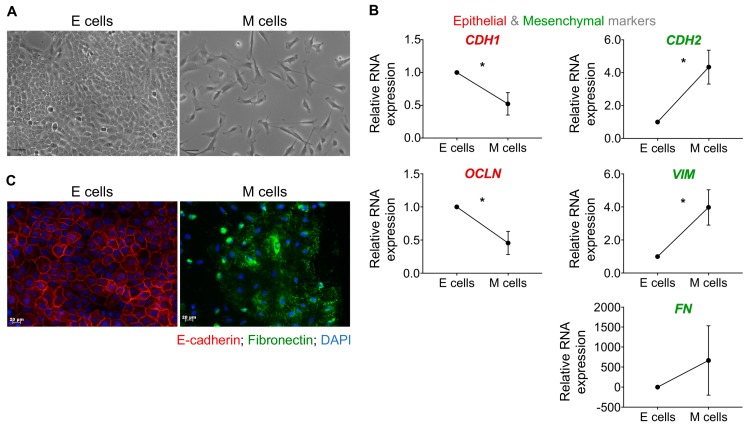
Characterization of a human Transforming Growth Factor beta TGFβ-induced epithelial to mesenchymal transition (EMT) cell model. (**A**) Bright-field microscopic images of epithelial and mesenchymal cells; (**B**) quantification of epithelial (*CDH1* and *OCLN*) and mesenchymal markers (*CDH2*, *VIM*, and *FN*) mRNA expression by qRT-PCR. Data obtained for M cells was normalized for E cells and *GAPDH* was used as endogenous control. Graphs represent the mean ± standard deviation of three independent experiments (**p* ≤ 0.05, unpaired t-test with Welch’s correction); and (**C**) immunofluorescence for E-cadherin (red staining), Fibronectin (green staining) in epithelial and mesenchymal cells. Scale bar: 20μm. (E—epithelial cells; M—mesenchymal cells).

**Figure 3 ijms-20-02608-f003:**
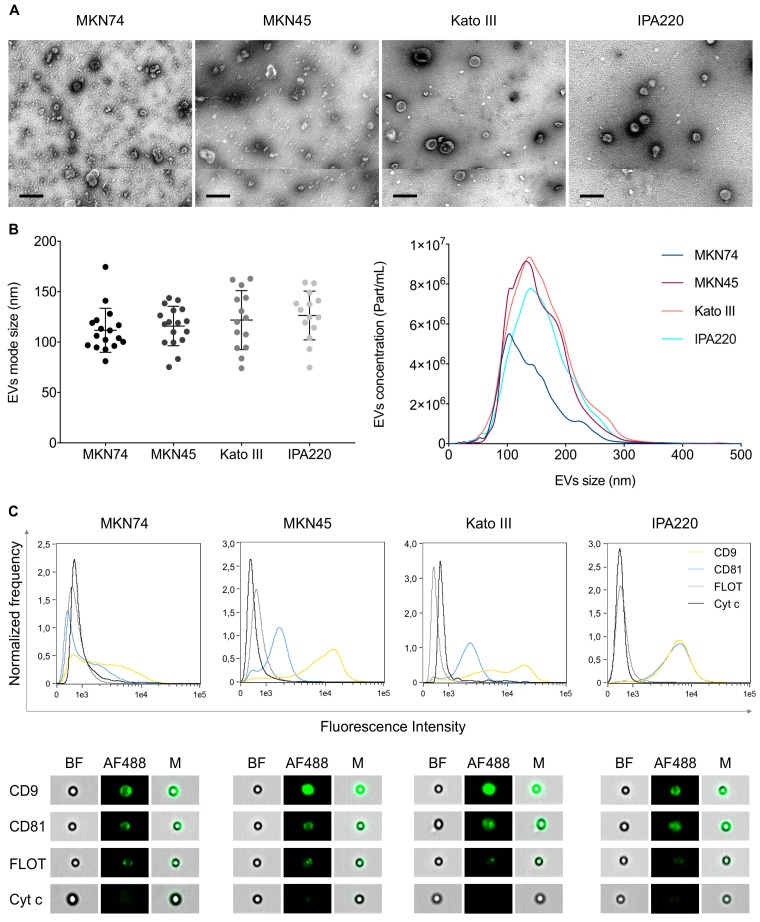
Characterization of EVs secreted by MKN74, MKN45, Kato III, and IPA220 GC cell lines. (**A**) Representative electron microscopy images of EVs isolated from GC cells. Scale bars: 200 nm; (**B**) NTA of isolated EVs with mode size distribution (left) and particle concentration (right). Graphs represent the mean ± standard deviation of at least 14 biological replicates; (**C**) detection of CD9, CD81, Flotillin-1, and Cytochrome C (negative control) by imaging flow cytometry. Distribution and representative images of the intensity of fluorescence detected for each marker in three biological replicates. Bright-field images (BF) show beads to which EVs were coupled, fluorescence images (AF488) show EVs labeled with specific markers, and merged images (M) show labeled EVs coupled to beads.

**Figure 4 ijms-20-02608-f004:**
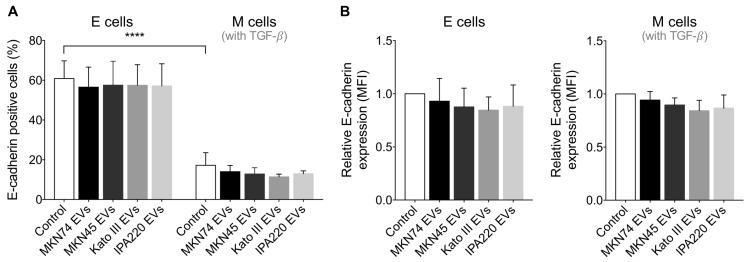
GC-EVs do not modulate membrane E-cadherin expression of epithelial and mesenchymal cells. (**A**) Percentage of E-cadherin-positive cells in non-treated (control) and treated epithelial and mesenchymal cells were assessed by flow cytometry. Graph represents the mean ± standard deviation of three independent experiments (**** *p* ≤ 0.0001, Two-way ANOVA with Tukey’s multiple comparisons test); (**B**) intensity of E-cadherin surface expression of non-treated (control) and treated epithelial and mesenchymal cells was assessed by flow cytometry. Graph represents the mean ± standard deviation of three independent experiments, and data were normalized to each respective control cell. TGFβ1 was maintained in M cultures (control and treated M cells).

**Figure 5 ijms-20-02608-f005:**
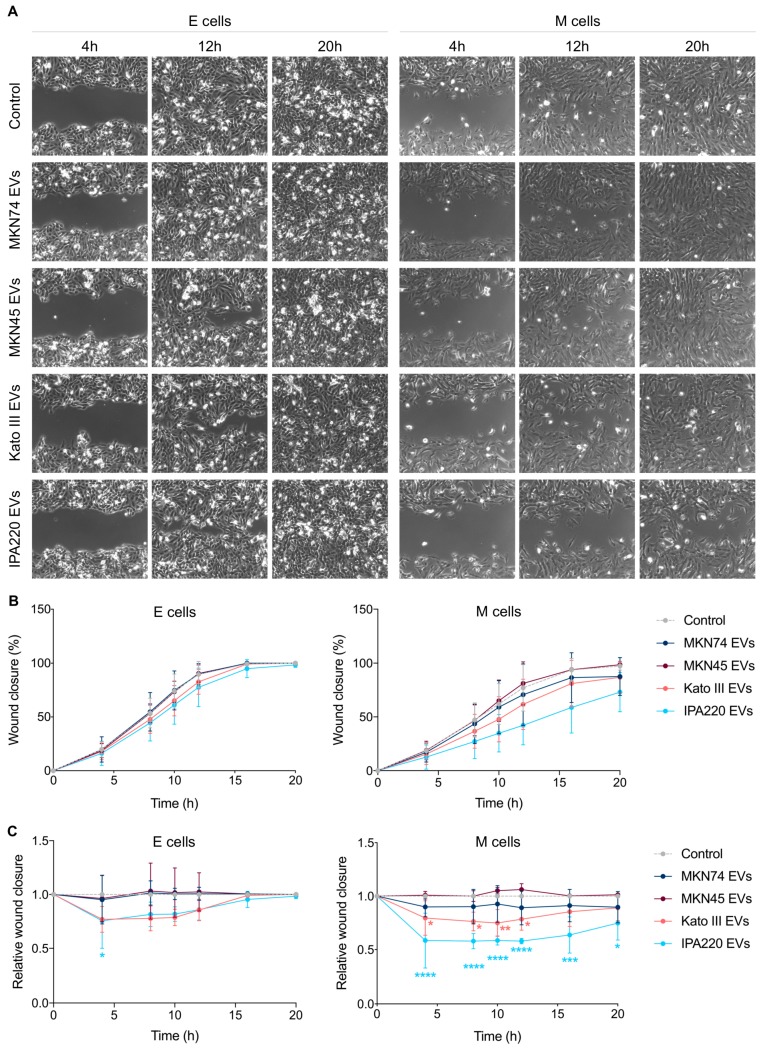
GC-EVs impair the migration of epithelial and mesenchymal cells. (**A**) Time-lapse microscopic images illustrating the migration of non-treated (control) and treated epithelial and mesenchymal cells, at distinct time-points; (**B**,**C**) the migration rate of non-treated (control) and treated epithelial and mesenchymal cells was assessed by time-lapse microscopy; (**B**) graphs represent the mean percentage of wound closure ± standard deviation of three independent experiments; (**C**) graphs represent the mean of normalized wound closure ± standard deviation of three independent experiments; the data was normalized to the control cells at each time-point (* *p* ≤ 0.05, ** *p* ≤ 0.01, *** *p* ≤ 0.001, **** *p* ≤ 0.0001, and two-way ANOVA with Dunnett’s multiple comparisons test; blue and orange asterisks correspond to E and M cells treated with IPA220-EVs or Kato III-EVs, respectively.

**Figure 6 ijms-20-02608-f006:**
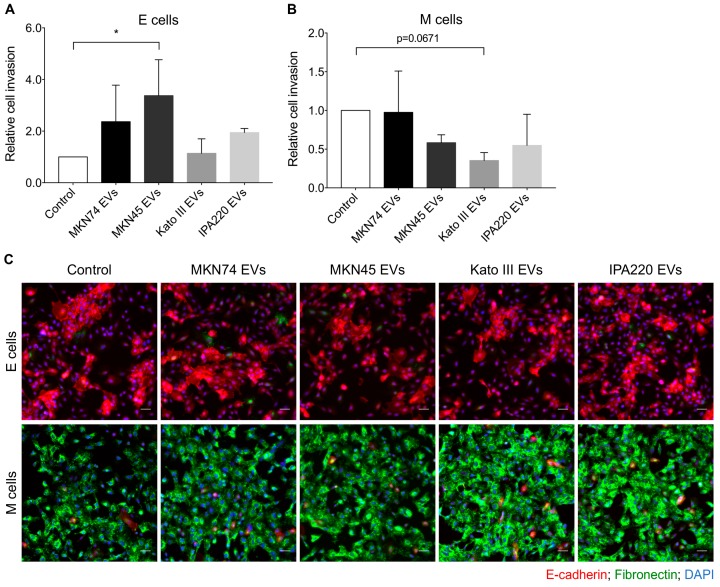
GC-EVs impair the invasion of epithelial and mesenchymal cells. The invasion levels of non-treated (control) and treated epithelial (**A**) or mesenchymal cells (**B**) were assessed by matrigel invasion assay; graph represents the mean ± standard deviation of three independent experiments and data was normalized to control cells (**p* ≤ 0.05, one-way ANOVA with Kruskal-Wallis test); (**C**) immunofluorescence for E-cadherin (red staining) and fibronectin (green staining) in non-treated (control) and treated epithelial and mesenchymal cells. Scale bars: 50μm.

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
