# Peer review of "Gastric Cancer Extracellular Vesicles Tune the Migration and Invasion of Epithelial and Mesenchymal Cells in a Histotype-Dependent Manner"

_ijms, 2019, doi:10.3390/ijms20112608_

Round 1
Reviewer 1 Report
This paper evaluated the effect of GC-EVs on the migration and invasion of epithelial cells and mesenchymal cells. The paper was very well written with methods and results clearly presented.
Below are some comments:
1. When evaluating the effect of GC-EVs on E-cadherin levels in both E and M cells, the author used the M cells and E cells derived from MCF10A with the presence of TGF-beta, then treated with/without GC-EVs. It will be more clinical relevant if compared E cells and T cells derived from MCF10A with/without GC-EV environment (TGF-beta ±GC-EV).
2. For the wound-healing migration assay and Matrigel invasion assay, normal (non-treated) E and M cells and cells treated with GC-EVs for 24hrs were used. Then both cells were cultured under normal conditions for 24 hrs before further experiment. Will these 24 hrs culture under normal conditions undermine the characteristics of GC-EV treated cells, thus affect the final results?
3. Please state why use the timepoint 24h for the E and M cell EV treatment.
4. The author did EV isolation through serial centrifugation and get a few big size (>200nm) EVs. According to our experience, the percentage of big size EVs will reduce when we do a high-speed centrifugation (10,000g for 30min) before ultracentrifugation to further move the debris/apoptotic bodies.
Author Response
Dear Professor Vincenza Dolo and Dr. Ilaria Giuti
Guest Editors
Special issue "Extracellular Vesicles and Metastatic Niche"
International Journal of Molecular Sciences
Porto, May 16th, 2019
Manuscript ID: ijms-491327
Title: Gastric Cancer Extracellular Vesicles Tune the Migration and Invasion of Epithelial and Mesenchymal Cells in a Histotype-dependent manner
We are deeply grateful to the Reviewers and Editors for their comments and suggestions to improve our manuscript. We have addressed every comment made to the manuscript and the responses are listed below, point-by-point. As requested by the Editor, we rephrased the highlighted sections of the Material and Methods section. We would like to mention that these sections have a high similarity with our own recently published paper (doi: 10.1002/advs.201800948), which was properly cited in the manuscript. Additionally, we included the requested changes in the manuscript (highlighted in yellow) and hope that this version fulfills the expectations of the Reviewers and Editors and meets the criteria to be accepted for publication in the International Journal of Molecular Sciences.
Looking forward to receiving your news.
Sincerely yours,
Sara Rocha, Sara Teles, Mafalda Azevedo, Patrícia Oliveira, Joana Carvalho and Carla Oliveira
Response to Reviewer 1 Comments
Point 1: “When evaluating the effect of GC-EVs on E-cadherin levels in both E and M cells, the author used the M cells and E cells derived from MCF10A with the presence of TGF-beta, then treated with/without GC-EVs. It will be more clinical relevant if compared E cells and M cells derived from MCF10A with/without GC-EV environment (TGF-beta ±GC-EV).”
Response 1: We appreciate the Reviewer’s concern and we would like to clarify that E cells were never exposed to TGFβ1 and that M cells can only be obtained by treating MCF10A epithelial cell line with TGFβ1 during seven days (detailed description can be found at the Material and Methods section: 4.1.1 Treatment of MCF10A epithelial cell line with TGFβ1). To keep the mesenchymal-like phenotype, M cells (control and GC-EVs treated) were always maintained in TGFβ1-supplemented medium, as stated in the Material and Methods section (“4.3.1 Treatment of Epithelial and Mesenchymal cells with GC-EVs” - lines 421-422). Neverthless, we reinforced this issue in the graphs of Figure 4 and by stating the following sentence in the legend of this figure (line 171): “…TGFβ1 was maintained in M cultures (control and treated M cells).
Point 2: “For the wound-healing migration assay and Matrigel invasion assay, normal (non-treated) E and M cells and cells treated with GC-EVs for 24hrs were used. Then both cells were cultured under normal conditions for 24 hrs before further experiment. Will these 24 hrs culture under normal conditions undermine the characteristics of GC-EV treated cells, thus affect the final results? Point 3. Please state why use the timepoint 24h for the E and M cell EV treatment.”
Responses 2 and 3: This is a highly relevant and challenging question; we thank the Reviewer for raising it. First of all, we would like to mention that these functional studies represent a proof of principle that GC-EVs influence the behavior of normal E and M cells in a histotype-dependent manner. We recognize that these particular points raised by Reviewer 1 constitute limitations of the study: 1) the culture of cells under normal conditions (i.e. cultured in EV-depleted medium), during 24h before functional assays, and; 2) the choice of a timepoint at 24h for treatments of E and M cells with GC-EVs. Regarding issue 1, it is important to state that controls and GC-treated cells were subjected to the same experimental conditions, thus we believe that the experimental setup is reliable as well as the results. Regarding issue 2, we chose the timepoint of 24h for the treatment of E and M cells with GC-EVs based on our previous reported results (doi: 10.1002/advs.201800948). In our previous work, we compared the invasion and proliferation of E cells treated with GC-EVs derived from 2D and 3D cultures, and realized that with the same experimental setup and the timepoint of 24h, it was possible to detect changes in these parameters. However, other timepoints would also be interesting to address in other more in-depth studies beyond our proof of principle study.
Point 4: “The author did EV isolation through serial centrifugation and get a few big size (>200nm) EVs. According to our experience, the percentage of big size EVs will reduce when we do a high-speed centrifugation (10,000g for 30min) before ultracentrifugation to further move the debris/apoptotic bodies.”
Response 4: We appreciate the Reviewer’s comment and, in fact, in the past our EV isolation protocol included the high-speed centrifugation step (10,000g for 30min). However, when we replaced this step by a 0.22mm filtration step, this change in the protocol allowed increasing the yield of recovered EVs without compromising their range size. In fact, these findings were supported by TEM analysis, which did not reveal the existence of cell debris or vesicles with sizes resembling apoptotic bodies. Moreover, this replacement of high-speed centrifugation step by filtration has been extensively used in the field (doi.org/10.3402/jev.v3.24858; doi.org/10.1016/j.ymeth.2015.05.028).

Reviewer 2 Report
Rocha et al. have analyzed the effect of gastric cancer extracellular vesicles in normal epithelial and mesenchymal cells (derived from epithelial cells after TGFB1 treatment). The hypothesis is interesting and in general the results and figures are presented properly. However, there are several questions that need to be answered before further decision on the present manuscript. The main concern is related to the normal cell line chosen for the study, an epithelial breast cell line.
Major points:
-The authors are working with extracellular vesicles (EV) obtained from gastric cancer cell lines and they would like to evaluate the effect of these EV in normal epithelial(E) and mesenchymal (M) cells surrounding the gastric tumor. However, since there are no available normal E gastric cell lines the authors used a breast cell line. I think this is an important limitation specially for the interpretation of the results. The authors should repeat the experiments either in primary E gastric cells or at least in a normal E cell line from the digestive system.
-I have been demonstrated that the EV fusion is mediated by the proteins present in their membrane and that EV have tropism trough different cell types. Then, not all cell lines have the same capacity to absorb specific EV. The authors have not evaluated whether the EV secreted by the gastric cell lines used were properly absorbed by the Epithelial and M cells used and if exists different grades of absorption depending on the cell of origin of the EV. This need to be evaluated in order to confirm the conclusions obtained.
-In some cell lines the migration result is not in accordance with the invasion result. This should be better discussed because is difficult to understand.
Minor points:
-The analysis of the WHA should be better explained in the methods section. How many distances analyze the authors? Or the authors analyzed the overall area?
-Figure 5B-C. The graph will be more easy to interpret if each cell line is represented in a different color, instead of using gray scale.
-The authors make their own EV-depleted medium. It is not indicated in the methods how the authors verify that the procedure was correctly performed. Did the authors analyzed the EV-depleted medium by Nanosight? For instance.
-How was established the quantity of EV necessary for each functional experiment? In some experiments the authors used 1x109 and in other experiment 5x108. Have the authors analyzed if a dose dependent effect can be appreciated?
Author Response
Dear Professor Vincenza Dolo and Dr. Ilaria Giuti
Guest Editors
Special issue "Extracellular Vesicles and Metastatic Niche"
International Journal of Molecular Sciences
Porto, May 16th, 2019
Manuscript ID: ijms-491327
Title: Gastric Cancer Extracellular Vesicles Tune the Migration and Invasion of Epithelial and Mesenchymal Cells in a Histotype-dependent manner
We are deeply grateful to the Reviewers and Editors for their comments and suggestions to improve our manuscript. We have addressed every comment made to the manuscript and the responses are listed below, point-by-point. As requested by the Editor, we rephrased the highlighted sections of the Material and Methods section. We would like to mention that these sections have a high similarity with our own recently published paper (doi: 10.1002/advs.201800948), which was properly cited in the manuscript. Additionally, we included the requested changes in the manuscript (highlighted in yellow) and hope that this version fulfills the expectations of the Reviewers and Editors and meets the criteria to be accepted for publication in the International Journal of Molecular Sciences.
Looking forward to receiving your news.
Sincerely yours,
Sara Rocha, Sara Teles, Mafalda Azevedo, Patrícia Oliveira, Joana Carvalho and Carla Oliveira
Response to Reviewer 2 Comments
Point 1: “The authors are working with extracellular vesicles (EV) obtained from gastric cancer cell lines and they would like to evaluate the effect of these EV in normal epithelial (E) and mesenchymal (M) cells surrounding the gastric tumor. However, since there are no available normal E gastric cell lines the authors used a breast cell line. I think this is an important limitation especially for the interpretation of the results. The authors should repeat the experiments either in primary E gastric cells or at least in a normal E cell line from the digestive system.”
Response 1: We appreciate and understand the Reviewer’s concern, which we also share. Unfortunately, primary epithelial gastric cells hardly proliferate and maintain viability in vitro. Therefore, the number of viable cells that we would be able to recover would be incompatible with the experimental design of our study (e.g. EMT induction time, number of cells for multiple GC-EVs treatments and functional assays). These limitations have led us to select a normal epithelial breast cell line, rather than the optimal gastric primary cells. Furthermore, we chose a breast epithelial normal cell line instead of a normal cell line from the digestive system, based on the following evidences:
1) there is a lack of reliable normal epithelial cell lines derived from the digestive system (e.g. colon cell line);
2) breast epithelial cancers, but not other cancers from the digestive system, have been identified in the tumour spectrum of GC-associated syndromes, such as Hereditary Diffuse Gastric Cancer (doi: 10.1001/jamaoncol.2014.168). This evidence supports breast epithelia as a tissue mimicking gastric epithelia regarding the tumorigenic process triggered by germline alterations.
Therefore, we believe that among the limited set of available normal epithelial cell lines, the normal epithelial breast cell line (MCF10A), although not ideal, is an acceptable model for our study.
Point 2: “I have been demonstrated that the EV fusion is mediated by the proteins present in their membrane and that EV have tropism through different cell types. Then, not all cell lines have the same capacity to absorb specific EV. The authors have not evaluated whether the EV secreted by the gastric cell lines used were properly absorbed by the Epithelial and M cells used and if exists different grades of absorption depending on the cell of origin of the EV. This need to be evaluated in order to confirm the conclusions obtained.”
Response 2: We thank and understand the Reviewer’s concern. In this study, we aimed at understanding how E and M recipient cells behave when challenged with EVs derived from different GC cell lines. We would like to reinforce that E and M cells treated with GC-EVs were never compared, so the difference in the capacity of absorption between E and M cells is not relevant for the conclusions of this manuscript. We agree that GC-EVs representative of different GC histotypes may have distinct membrane compositions, which may influence their uptake by E and M recipient cells. In fact, potential distinctive uptakes of GC-EVs may also inform on how effectively these EVs may convey tumor messages to its microenvironment. For that reason, we chose to normalize the effects observed on recipient cells to the number of EVs used in the treatment rather, than to their uptake. Nevertheless, we would like to mention that in our previous study (doi: 10.1002/advs.201800948), we observed that MCF10A cells (which represent only E cells) internalized EVs derived from MKN45 and MKN74 GC cells with similar efficiencies. Therefore, we believe that our results translate the real capacity of modulation by EVs and not a technical issue related with distinct uptake efficiencies.
Point 3: “In some cell lines the migration result is not in accordance with the invasion result. This should be better discussed because is difficult to understand.”
Response 3: We thank the Reviewer’s for raising this point. Indeed, we would should better clarify that cell migration and invasion represent two distinct and quite independent biological processes. Whereas migration refers to the direct movement of cells on 2D surfaces, such as plastic plates, without an obstructive fiber network, invasion refers to the active movement of cells through a 3D matrix, such as matrigel, which is accompanied by degradation of the 3D environment. Therefore, it was interesting for us to obtain results that support the differences in these two processes, when epithelial normal cells are challenged by GC-EVs. Indeed, in some cell lines the migration and invasion results are effectively not concordant.
Point 4: “The analysis of the WHA should be better explained in the methods section. How many distances analyze the authors? Or the authors analyzed the overall area?”
Response 4: We thank and agree with the Reviewer’s comment. We analyzed the overall area of the wound. We have rephrased the WHA section in the material and methods and explained better the analysis of wound-healing (line 438-449): “Wound-healing assays of E and M cells (treated and non-treated with GC-EVs) were performed using µ-Plate 24 well with culture-inserts (Ibidi) and time-lapse microscopy. Briefly, 4x104 cells were seeded in each of the two wells of the culture insert and grown to confluence for, approximately, 24h. After this period, the culture insert was removed, cells were washed with PBS and fresh EV-depleted medium supplemented with GC-EVs (5x108) was added. Bright field images of the wounds were photographed immediately after removing the insert (0h) and again at 4h, 8h, 10h, 12h, 16h and 20h using Leica DMI6000. Five images were automatically captured per well per time point to completely image each wound. Cells were maintained at 37 °C in 5% CO2 humidified atmosphere during the time-lapse imaging process. The total area of the wound was measured at each time point using ImageJ software (MRI_Wound_Healing_Tool). Results represent the mean ± standard deviation of three biological replicates. Data was analyzed with a two-way ANOVA with Dunnett’s multiple comparison test.”
Point 5: “Figure 5B-C. The graph will be more easy to interpret if each cell line is represented in a different color, instead of using gray scale.”
Response 5: In compliance with Reviewer’s suggestion, we have represented the graphs (figure 5B-C) with a different color for each cell line.
Point 6: “The authors make their own EV-depleted medium. It is not indicated in the methods how the authors verify that the procedure was correctly performed. Did the authors analyzed the EV-depleted medium by Nanosight? For instance.”
Response 6: We appreciate the Reviewer’s concern and we would like to mention that we analyzed the EV-depleted medium by TEM, which confirmed the absence of vesicles (Please see figure below). This analysis is now stated in the material and methods section (lines 374-375): “EV-depleted medium was further submitted to TEM to ascertain that the depletion was correctly performed.”
Point 7: “How was established the quantity of EV necessary for each functional experiment? In some experiments the authors used 1x109 and in other experiment 5x108. Have the authors analyzed if a dose dependent effect can be appreciated?”
Response 7: We thank the Reviewer’s question. The quantity of EVs necessary for functional assays was based on our previous results (doi: 10.1002/advs.201800948), in which we have evaluated the invasion and proliferation capacity of MCF10A cells treated with MKN45 and MKN74 EVs derived from 2D and 3D cultures. In the present study, we first performed a general treatment of E and M cells with 1x109 GC-EVs, and then submitted the treated cells to proliferation, invasion and migration assays. Specifically, for the migration assays, we performed a second treatment of E and M cells with 5x108 EVs, a quantity adapted to the lower number of cells used in this assay.

Reviewer 3 Report
The work presented by Oliveira et al aimed at demonstrate the role of GC-EVs in migration and invasion of epithelial and mesenchymal cells from tutor microenvironment. This study is interesting however there is a lack of in vivo results to further support the proposed findings. Thus authors need to provide some more explanations:
1)Authors must perform: Western Blot analysis to detect characteristic biomarkers enriched in EV populations using proper negative controls to exclude the presence of other intra-citoplasmatic contaminants (https://doi.org/10.1080/20013078.2018.1535750)
2) Authors claim that GC-EVs modulate migration and invasion of epithelial and mesenchymal cells. Which kind of mechanism is expected? Transcriptomic analysis would be of help
3) In Materials and Methods a proper section for Statistical Analysis is missing
Author Response
Dear Professor Vincenza Dolo and Dr. Ilaria Giuti
Guest Editors
Special issue "Extracellular Vesicles and Metastatic Niche"
International Journal of Molecular Sciences
Porto, May 16th, 2019
Manuscript ID: ijms-491327
Title: Gastric Cancer Extracellular Vesicles Tune the Migration and Invasion of Epithelial and Mesenchymal Cells in a Histotype-dependent manner
We are deeply grateful to the Reviewers and Editors for their comments and suggestions to improve our manuscript. We have addressed every comment made to the manuscript and the responses are listed below, point-by-point. As requested by the Editor, we rephrased the highlighted sections of the Material and Methods section. We would like to mention that these sections have a high similarity with our own recently published paper (doi: 10.1002/advs.201800948), which was properly cited in the manuscript. Additionally, we included the requested changes in the manuscript (highlighted in yellow) and hope that this version fulfills the expectations of the Reviewers and Editors and meets the criteria to be accepted for publication in the International Journal of Molecular Sciences.
Looking forward to receiving your news.
Sincerely yours,
Sara Rocha, Sara Teles, Mafalda Azevedo, Patrícia Oliveira, Joana Carvalho and Carla Oliveira
Response to Reviewer 3 Comments
Point 1: “Authors must perform: Western Blot analysis to detect characteristic biomarkers enriched in EV populations using proper negative controls to exclude the presence of other intra-citoplasmatic contaminants (https://doi.org/10.1080/20013078.2018.1535750).”
Response 1: We appreciate and thank the Reviewer’s suggestion. We chose imaging-flow cytometry to detect commonly associated exosomal markers CD9, CD81, and Flotillin-1, instead of western-blot, as it represents a more sensitive technique, when dealing with samples from which low amount of material can be recovered. We agree that a negative control should be included to exclude the presence of other intra-citoplasmatic contaminants, and we chose Cytochrome C as a negative control. These data is now included in Figure 3C.
Point 2: “Authors claim that GC-EVs modulate migration and invasion of epithelial and mesenchymal cells. Which kind of mechanism is expected? Transcriptomic analysis would be of help”
Response 2: We thank the Reviewer and find this suggestion very relevant (please see line 286-295 of the discussion). In our previous paper (doi: 10.1002/advs.201800948), we performed small-RNA sequencing and proteomics analysis of EVs derived from MKN45 and MKN74 cells, and the results were quite interesting. It is in our future plans to characterize not only EVs derived from Kato III and IPA220 cells, but also M and E recipient cells upon treatments with specific GC-EVs.
Point 3: “In Materials and Methods a proper section for Statistical Analysis is missing”
Response 3: Following the Reviewer’s suggestion, we added a section of statistical analysis in the Material and Methods (line 473-478): “4.4 Statistical Analysis - The Unpaired t test with Welch’s correction was used for comparisons of RNA expression levels of epithelial and mesenchymal markers. Flow-cytometry and wound-healing data were analyzed by a Two-way ANOVA with Tukey’s and Dunnett’s multiple comparisons tests, respectively. Matrigel invasion data was analyzed with a One-way ANOVA with Kruskal-Wallis test.”

Round 2
Reviewer 2 Report
The authors have answered most of my raised questions. However, there are some information that still need to be included in the discussion of the paper.
Previous POINT 1: The authors stated that "there is a lack of reliable normal epithelial cell lines derived from the digestive system (e.g. colon cell line);". However several normal colon cell lines are available at ATCC (https://www.atcc.org/~/media/PDFs/Cancer%20and%20Normal%20cell%20lines%20tables/Colon%20cancer%20and%20normal%20cell%20lines.ashx). Please, at least include a comment on the the discussion about this limitation of the study of suing breast normal cells and why you consider that is a valid model.
Previous POINT 2: The authors stated that "We would like to reinforce that E and M cells treated with GC-EVs were never compared, so the difference in the capacity of absorption between E and M cells is not relevant for the conclusions of this manuscript." In my opinion its relevant, since one of your conclusion is related with differences observed according the cell linage of the cell line og origin of the microvesicles. Please discuss this in the discussion.
Previous POINT 3: I agree with the authors that migration and invasion are different, but close mechanisms. However, I think that this point need to be discussed in the paper.
Author Response
Dear Professor Vincenza Dolo and Dr. Ilaria Giuti
Guest Editors
Special issue "Extracellular Vesicles and Metastatic Niche"
International Journal of Molecular Sciences
Porto, May 22nd, 2019
Manuscript ID: ijms-491327
Title: Gastric Cancer Extracellular Vesicles Tune the Migration and Invasion of Epithelial and Mesenchymal Cells in a Histotype-dependent manner
We are deeply grateful to the Reviewers and Editors for their comments and suggestions to improve our manuscript. We have addressed every comment made to the manuscript and included the requested changes (highlighted in yellow). We hope that this version fulfills the expectations of the Reviewers and Editors and meets the criteria to be accepted for publication in the International Journal of Molecular Sciences.
Looking forward to receiving your news.
Sincerely yours,
Sara Rocha, Sara Teles, Mafalda Azevedo, Patrícia Oliveira, Joana Carvalho and Carla Oliveira
Response to Reviewer 2 Comments
Previous POINT 1: The authors stated that "there is a lack of reliable normal epithelial cell lines derived from the digestive system (e.g. colon cell line);". However several normal colon cell lines are available at ATCC(https://www.atcc.org/~/media/PDFs/Cancer%20and%20Normal%20cell%20lines%20tables/Colon%20cancer%20and%20normal%20cell%20lines.ashx). Please, at least include a comment on the discussion about this limitation of the study of suing breast normal cells and why you consider that is a valid model.
Response to Previous POINT 1: We thank the Reviewer’s suggestion. We noticed that ATCC commercializes normal colon cell lines, although our collaborators working on Colorectal Cancer mentioned that these cell lines are not reliable. We followed the Reviewer’s suggestion by including a comment, in the discussion, regarding the usage of a normal breast cell line (lines: 272-282):”... We chose to use the non-tumorigenic breast epithelial cell line (MCF10A) as a recipient cell type to induce EMT given that: 1) non-tumorigenic gastric cell lines are currently not available; 2) primary epithelial gastric cells hardly proliferate and maintain viability in vitro, which make them incompatible with our experimental design; 3) breast epithelial cancers, but rarely other cancers from the digestive system, have been identified in the tumour spectrum of GC-associated syndromes, such as Hereditary Diffuse Gastric Cancer [33]. This last evidence supports breast epithelia as a tissue mimicking gastric epithelia regarding the tumorigenic process triggered by germline alterations. Altogether, the normal epithelial breast cell line (MCF10A), although not ideal, seems like an acceptable model to be used as a recipient cell line for GC EVs. This recipient model underwent EMT when treated with TGFβ1...”
Previous POINT 2: The authors stated that "We would like to reinforce that E and M cells treated with GC-EVs were never compared, so the difference in the capacity of absorption between E and M cells is not relevant for the conclusions of this manuscript." In my opinion its relevant, since one of your conclusion is related with differences observed according the cell linage of the cell line of origin of the microvesicles. Please discuss this in the discussion.
Response to Previous POINT 2: We thank and understand the Reviewer’s concern. Therefore, we included a comment on the potential distinct uptake in the discussion section (lines: 324-328):”… These results suggest that GC-EVs representative of different GC histotypes may have distinct cargo and membrane compositions, which may influence their uptake by recipient cells. On the other hand, epithelial (E) and mesenchymal (M) cells may also uptake and perceive GC-EVs differently. In fact, potential distinctive uptakes of GC-EVs by recipient cells may represent how effectively these EVs may convey tumor messages to its microenvironment.”
Previous POINT 3: I agree with the authors that migration and invasion are different, but close mechanisms. However, I think that this point need to be discussed in the paper.
Response to Previous POINT 3: We followed the Reviewer’s suggestion and included the following comment on the discussion (lines: 315-317): “…Interestingly, this invasive phenotype was not accompanied by an increased migration. As so, one may hypothesize that the cargo of MKN45-EVs is more important for the remodeling of the basement membrane than for the direct movement of epithelial cells.”
Reviewer 3 Report
Authors improved the manuscript which is now suitable for publication
Author Response
Dear Professor Vincenza Dolo and Dr. Ilaria Giuti
Guest Editors
Special issue "Extracellular Vesicles and Metastatic Niche"
International Journal of Molecular Sciences
Porto, May 22nd, 2019
Manuscript ID: ijms-491327
Title: Gastric Cancer Extracellular Vesicles Tune the Migration and Invasion of Epithelial and Mesenchymal Cells in a Histotype-dependent manner
We are deeply grateful to the Reviewers and Editors for their comments and suggestions to improve our manuscript. We have addressed every comment made to the manuscript and included the requested changes (highlighted in yellow). We hope that this version fulfills the expectations of the Reviewers and Editors and meets the criteria to be accepted for publication in the International Journal of Molecular Sciences.
Looking forward to receiving your news.
Sincerely yours,
Sara Rocha, Sara Teles, Mafalda Azevedo, Patrícia Oliveira, Joana Carvalho and Carla Oliveira
Response to Reviewer 3 Comments
Authors improved the manuscript which is now suitable for publication
We would like to acknowledge Reviewer 3 for the previous suggestions that truly improved our manuscript.